# Development of an Image Analysis-Based Prognosis Score Using Google’s Teachable Machine in Melanoma

**DOI:** 10.3390/cancers14092243

**Published:** 2022-04-29

**Authors:** Stephan Forchhammer, Amar Abu-Ghazaleh, Gisela Metzler, Claus Garbe, Thomas Eigentler

**Affiliations:** 1Eberhardt Karls Universität, Universitäts-Hautklinik, 72076 Tübingen, Germany; amar.abu-ghazaleh@student.uni-tuebingen.de (A.A.-G.); claus.garbe@med.uni-tuebingen.de (C.G.); 2Zentrum für Dermatohistologie und Oralpathologie Tübingen/Würzburg, 72072 Tübingen, Germany; metzler@zentrum-dermatohistologie.de; 3Department of Dermatology, Venereology and Allergology, Charité—Universitätsmedizin Berlin, Corporate Member of Freie Universität Berlin and Humboldt-Universität zu Berlin, Luisenstrasse 2, 10177 Berlin, Germany; thomas.eigentler@charite.de

**Keywords:** melanoma, prognosis, risk score, deep learning, artificial intelligence, Google’s teachable machines

## Abstract

**Simple Summary:**

The increase in adjuvant treatment of melanoma patients makes it necessary to provide the most accurate prognostic assessment possible, even at early stages of the disease. Although conventional risk stratification correctly identifies most patients in need of adjuvant treatment, there are some patients who, despite having a low tumor stage, have poor prognosis and could therefore benefit from early therapy. To close this gap in prognosis estimation, deep learning-based image analyses of histological sections could play a central role in the future. The aim of this study was to investigate whether such an analysis is possible only using basic image analysis of 831 H&E-stained melanoma sections using Google’s Teachable Machine. Although the classification obtained does not provide an additional prognostic estimate to conventional melanoma classification, this study shows that prognostic prediction is possible at the mere cellular image level.

**Abstract:**

Background: The increasing number of melanoma patients makes it necessary to establish new strategies for prognosis assessment to ensure follow-up care. Deep-learning-based image analysis of primary melanoma could be a future component of risk stratification. Objectives: To develop a risk score for overall survival based on image analysis through artificial intelligence (AI) and validate it in a test cohort. Methods: Hematoxylin and eosin (H&E) stained sections of 831 melanomas, diagnosed from 2012–2015 were photographed and used to perform deep-learning-based group classification. For this purpose, the freely available software of Google’s teachable machine was used. Five hundred patient sections were used as the training cohort, and 331 sections served as the test cohort. Results: Using Google’s Teachable Machine, a prognosis score for overall survival could be developed that achieved a statistically significant prognosis estimate with an AUC of 0.694 in a ROC analysis based solely on image sections of approximately 250 × 250 µm. The prognosis group “low-risk” (*n* = 230) showed an overall survival rate of 93%, whereas the prognosis group “high-risk” (*n* = 101) showed an overall survival rate of 77.2%. Conclusions: The study supports the possibility of using deep learning-based classification systems for risk stratification in melanoma. The AI assessment used in this study provides a significant risk estimate in melanoma, but it does not considerably improve the existing risk classification based on the TNM classification.

## 1. Introduction

Over the past years and decades, there has been a significant increase in the incidence of malignant melanoma [1]. Despite major advances in the treatment of metastatic melanoma, including targeted therapy with BRAF inhibitors or immune checkpoint blockade, malignant melanoma remains the skin tumor responsible for the highest number of skin tumor-associated deaths worldwide, with approximately 55,500 cases [2]. Histologically, different subtypes of malignant melanoma can be distinguished. According to the currently valid World Health Organization Classification published in 2018, a distinction is made between melanomas that typically occur in chronically sun-damaged (CSD) skin and those that typically do not occur in chronically sun-damaged skin. These differ in the underlying genetic pathways. The most common subtypes, superficial spreading melanoma (low-CSD) and lentigo maligna melanoma (high-CSD), but also desmoplastic melanoma, are found in association with sun-damaged skin. Representatives of melanomas that do not occur in chronically sun-damaged skin (no-CSD) are acral-, mucosal- and uveal-melanomas, Spitz melanomas, melanomas originating from congenital nevi or blue nevi. Nodular melanomas, on the other hand, can be found in both groups with different underlying genetic pathways [3]. Prognosis prediction and staging of melanoma are mainly based on histologic diagnosis in primary tumors. In this context, tumor thickness (according to Breslow) and ulceration are included in the 8th AJCC classification [4]. In the case of additional histological features, such as regression and mitotic rate, an impact on the further prognosis is assured [5,6,7]. Other prognostic factors result from the primary staging diagnosis, which includes sonography, CT section imaging and sentinel node biopsy depending on the stage [4,8]. With the advent of adjuvant therapy options for patients with high-risk tumors, the most accurate prognostic prediction possible is already necessary for the primary tumor. Since adjuvant immune checkpoint therapy has a non-negligible side effect profile, it is crucial to identify patients who may particularly benefit from such therapy. Various gene-expression-based assays are in development to distinguish high-risk patients from those with only low risk of metastasis [9,10,11,12]. However, these studies are cost intensive and therefore cannot yet be widely used. In addition, these examinations consume tissue that may be needed for further diagnostic workup. The morphology of melanoma already shows a very high diversity in the H&E section, which goes far beyond the detection of tumor thickness, ulceration, mitotic rate and regression. A grading, which is common for most other tumor types, such as cutaneous squamous cell carcinoma, does not exist for melanoma. With the onset of digitalization in pathology, artificial intelligence (AI)-based image analysis has created new opportunities in the evaluation of histological sections. AI is a set of technologies that enables computer systems to acquire intelligent capabilities. One branch of AI is the concept of machine learning, which gives computers the ability to learn without being explicitly programmed [13]. Deep learning, which is popular today, is characterized by greater network depth in terms of multiple layers of neurons; therefore, it makes it possible to learn and solve even complex tasks. Remarkably, artificial neural networks make this possible without having to deposit specific rules or instructions beforehand [14]. It has been shown that programs based on artificial intelligence are able to achieve high diagnostic accuracy in diagnosing melanoma from dermatoscopic images [15,16,17,18]. Diagnosis by artificial neuronal networks also seems to lead to very reliable results for histological sections. For epithelial skin tumors, but especially for melanomas, programs have been developed that enable robust diagnostics [19,20,21,22,23,24]. More exciting, though, is the question of whether image analysis with artificial neural networks can not only confirm a diagnosis but whether it is conceivable that subvisual structures or patterns in histological sections can be detected, leading to improved prognosis assessment. First studies in melanoma have shown that it may be possible to achieve prognosis prediction, prediction of sentinel positivity and prediction of response to immunotherapy by using artificial intelligence-assisted image analysis [25,26,27]. In particular, the work of Kulkarni et al. was able to make an impressive prognosis prediction based on image analysis; however, here a complex algorithm was used which, in addition to the mere morphological tumor cell information, evaluates in part the distribution of inflammatory cells [26]. Since a clear impact on melanoma prognosis has been well studied, especially for tumor-infiltrating lymphocytes, it remains unclear whether a purely morphological image analysis of tumor cells allows melanoma prognosis [28,29,30].

The aim of our study was to develop a prognosis score based purely on histological photographs to predict survival in melanoma. Since our score should be made publicly available, easy to use and based solely on morphological image information, Google’s Teachable Machine was used as a deep learning program. This is a pre-trained neural network for image analysis that allows the classification of images into certain groups after previous training [31]. Google’s Teachable Machine uses the basic framework of TensorFlow. This is a platform released in 2015 that was created to make artificial intelligence and its training accessible to the public. The use of this program has already been investigated in the first studies for image analysis of medical questions [32].

## 2. Materials and Methods

### 2.1. Study Population

All 2223 patients diagnosed with primary melanoma at the University Dermatological Clinic Tübingen between 1 January 2012 and 31 December 2015 who provided written informed consent to the nationwide melanoma registry were included in the study. All 831 patients with follow-up data of at least 2 years and histological sections in our archive were included in the further analysis. The group “dead” consists of all patients that died due to melanoma during the observation period up to 114 months. The group “alive” consists of all patients that were alive, lost to follow up or died of another reason. Alive patients with follow-up for less than 2 years were excluded from the study. The diagnosis of melanoma was made by at least two experienced, board-certified dermatopathologists (SF, GM).

### 2.2. Digitization of HE Sections and AI-Based Evaluation

All H&E sections of primary melanoma were photographed at the site of the highest tumor thickness according to Breslow using 100× magnification (Figure 1). Pictures were taken using a Nikon Eclipse 80i microscope mounted with a Nikon Digital Sight DS-FI2 camera. The program Nikon NIS Elements D Version 4.13.04. was used, and the exposure time was set to 3 ms. The data were saved in JPG format. Images were analyzed using Google’s Teachable Machine, a pre-trained neural network [31]. Sixty percent of the 831 images served as the training cohort, and 40% of the images were subsequently evaluated as the test cohort. The allocation of the 500 images to the training cohort or the 331 images to the test cohort was random. The training dataset contains images that were only used for training Google’s Teachable Machine. An analysis of these data was not performed later. Of these 500 patients, 429 were alive; thus, these images were used for the training of the “alive” group. Of the 500 patients, 71 were deceased; these were used for the training of the group “dead”. The training was carried out twice and separately for the groups “whole images” and “area of interest”. The training curves for accuracy and loss were obtained for both training groups and are shown in Appendix A. The model that emerged from the initial training was used for further assessment. As Google’s Teachable Machine does not provide a verification function, a separate set of verification data was not assigned. The remaining 331 patients were used as the test cohort. The images of these patients were not previously seen by Google’s Teachable Machine. These 331 patient images were then classified by the program into the categories “dead” and “alive”. Patients who were classified as “dead” were given the label “high-risk” in the further study, and patients who were classified as “alive” were given the label “low-risk”.

The evaluations of the whole images or the “area of interest” images were performed separately. During the evaluation of whole images, the uploaded images in landscape format 4:3 were cut by Google’s Teachable Machine into a square format. To balance the training groups “alive” and “dead”, the images of the group “dead” were used 6 times.

For the “area of interest” evaluation, representative image sections of about 250 × 250 µm were selected from the images by a dermatopathologist showing representative tumor areas (file size from 103 kB to 622 kB). Whenever possible, we selected representative areas from the dermal tumor compartment. Only in cases with a very small tumor thickness were areas with an epidermal component included (see Figure 1). To balance the training groups “alive” and “dead” the images of the group “dead” were cut into 6 representative tumor areas. In the advanced settings of the “Teachable Machine” the epochs were set to 1000, the batch size to 16 and the learning rate to 0.001. The 334 images of the test cohort were uploaded individually, and the group allocation of Google’s Teachable Machine and the indicated percentage were collected.

### 2.3. Statistics

Statistical calculations were performed using IBM SPSS Statistics Version 23.0 (IBM SPSS, Chicago, IL, USA). Numerical variables were described by mean value and standard deviation or median values and interquartile range (IQR). Receiver operating characteristic (ROC) curve analyses and corresponding *p*-value calculations were performed using the ROC-Analysis tool in SPSS. *p*-values in Kaplan–Meier curves were calculated using the log-rank (Mantel-Cox) test. Throughout the analysis, *p* values < 0.05 were considered statistically significant.

## 3. Results

To create a prognosis score for melanoma, 60% (*n* = 500) of the images were used as a training cohort. For this purpose, the images were categorized as “alive” and “dead”, according to the actual survival of the patients. Google’s Teachable Machine was used to create an algorithm from these training groups, which was then applied to the test cohort. The training curves of the models showed an overfitting (see Appendix A); therefore, the training was repeated with a new randomized training set to avoid possible bias caused by the grouping (Appendix A). Since the repetition also showed comparable overfitting, the evaluation was continued with the initial trained model. Subsequently, the remaining 40% of the images (*n* = 331) were used as a test of the previously created score. The overall cohort had a median age of 62 years at diagnosis, a preponderance of 55.6% men versus 44.4% women, and a median tumor thickness of 1.05 mm. Ulceration was detectable in 21.3% of the patients. The most common histological subtype was superficial spreading melanoma with 59.3%, followed by nodular melanoma with 16.1%, lentigo maligna melanoma with 9.1%, acrolentiginous melanoma with 6.0%, other melanomas (5.7%) and melanomas of an unknown subtype (3.5%). Most melanomas were found on the trunk (41.4%), followed by melanomas of the lower extremity (26.4%), head and neck (17.7%), and upper extremity (14.1%). At initial diagnosis, 64.3% of patients were classified as stage I, 21% as stage II, 13.4% as stage III, and 1.3% as stage IV. The staging, subtype classification, and epidemiologic data showed comparable values in the training and test cohorts, confirming the randomization of the groups (see Table 1).

Figure 1 shows the procedure for photographing the melanoma sections. In many melanomas, tumors were present in numerous blocks and slides. The H&E section with the highest tumor thickness according to Breslow was selected (see Figure 1a). Here, an image was taken at 100× magnification at the site of the highest tumor thickness. This image was used for the “whole image” analysis. From these “whole images”, small image sections (about 250 × 250 µm) were selected that showed representative parts of the tumor. The generation of a prognosis score was initially performed on both groups. These were compared by ROC analysis (see Figure 2a). We investigated how reliably a prognostic prediction of overall survival could be made based only on the AI classifier. When analyzing the “whole images”, no significant result (*p* = 0.101) could be obtained in the prediction of overall survival. The classifier showed an AUC of 0.581, which was only slightly better than a random classification (AUC of 0.5). In contrast, however, a significant prediction estimate with an AUC of 0.694 (*p* < 0.001) could be obtained with the analysis of the AOI images. Therefore, further evaluation was performed using the classifier generated by the analysis of the area of interest images.

If one only uses the classifier, generated solely by image analysis of a H&E-stained melanoma section, this already allows a good prognosis estimate of the overall survival. Of the 331 patients in the test cohort, 230 patients were assigned the AI-classifier “low-risk” and 101 patients were given the AI-classifier “high-risk”. Malignant melanoma-related overall survival was 88.2% in the test cohort, with 39 deaths in the observation period up to 114 months. The AI-classifier “low-risk” group showed a statistically significant better overall survival of 93% with 16 deaths, compared to a survival of 77.2% and 23 deaths in the AI-classifier “high-risk” group (*p* < 0.001). Figure 3a shows the Kaplan–Meyer survival curves of the total test cohort, related to melanoma-specific overall survival. Considering recurrence-free survival, there is also a statistically significant distinction by grouping into AI-classifier “low-risk” and “high-risk” (*p* < 0.001). Of the 230 patients in the “low-risk” group, an event such as recurrence, metastasis or death from the disease was recorded in 43 cases. This leads to a recurrence-free survival rate of 81.3%. In contrast, 37 events were recorded in the AI-classifier “high-risk” group out of 101 patients, resulting in a recurrence-free survival of only 63.4% (Figure 3b).

Next, we questioned whether the AI classifier could complement the existing forecast prediction with the AJCC 2017 classification. Here, we first performed a ROC analysis. Comparing the prognosis estimate resulting from the existing T-classification (according to AJCC 2017) of the primary tumor (tumor thickness according to Breslow and the presence of an ulceration) (AUC = 0.872) with the prognosis estimate resulting from the addition of the AI-classifier (AUC = 0.881), only a slightly improved risk stratification was shown (see Figure 2b). This was also evident in the analysis of the Kaplan–Meyer curves of overall survival for the individual stages of the AJCC 2017 classification. Looking at AI-based risk classification in stage I, the following picture emerges: of the 207 patients in Stage I of the test cohort, 163 (79%) received the label AI-classifier “low-risk”. Of these 163 patients, 2 died during the observation period, corresponding to an overall survival rate of 98.8%. Forty-four patients (21%) were classified as “high-risk”. In this group, there were also two deaths, which corresponds to an overall survival rate of 95.5%. However, with a *p*-value of 0.154, this does not reach statistical significance (Figure 4a). Regarding stage II, of 74 patients, 39 (53%) were classified as “low-risk,” and 35 patients (47%) were marked as “high-risk.” There were 8 deaths in the “low-risk” group resulting in an overall survival of 79.5%. The “high-risk” group had 10 deaths, resulting in an overall survival of 71.4%. However, this difference did not reach statistical significance with a *p*-value of 0.378 (Figure 3b). Stage III demonstrated the clearest differences in prognosis estimation. In our test cohort, there were 43 patients in stage III, of which 11 patients died during the observation period, resulting in an overall survival of 74.4%. Twenty-five of these patients (58%) were considered “low-risk”, and in fact, only 4 deaths occurred in this group, resulting in an overall survival of 84%. Of the 18 patients (42%) designated as “high-risk” by the AI-classifier, 7 patients died, resulting in an overall survival of only 61.1%. Although an early and quite clear separation of the Kaplan–Meyer curves is seen in stage III, no statistically significant difference (*p* = 0.159) results due to the rather small number of cases in this group (Figure 4c). Seven patients were found to be stage IV at initial diagnosis. Four of these were identified as AI-classifier “high-risk” and 3 were classified as “low-risk”. All patients in the “high-risk” group died during the observation period, resulting in an overall survival of 0%. In the “low-risk” group, 2 melanoma-specific deaths were recorded, resulting in a melanoma-specific survival of 33.3%. Patients in the “high-risk” group, in contrast, survived longer than those in the “low-risk” group. This leads to a statistically significant difference in the group classification at this stage (*p* = 0.018) (Figure 4d).

## 4. Discussion

### 4.1. Results

The present study demonstrates the possibilities offered using deep learning-based image analysis in the risk stratification of melanoma. Although the program for risk assessment merely has a tiny image of about 250 × 250 µm at its disposal and no further information is available, a quite reliable and statistically significant risk stratification can be achieved. However, the AI classifier used here does not significantly improve the existing risk classification based on the TNM classification. Nevertheless, it seems possible that such a classifier may add prognostic value to conventional prognostic factors. In particular, our survival data in stage III show a tendency toward improved prognosis with the addition of the AI-classifier, even if this does not reach statistical significance. Further studies with a larger cohort from this advanced tumor stage are needed to confirm this.

The first published studies have investigated the use of AI-based neural networks in melanoma. It has been shown that such image analysis can reliably detect melanomas and differentiate them from benign melanocytic nevi [19,20,21,22]. Predicting prognosis, though, is much more complex than mere diagnostic classification of nevus and melanoma. Hence, a study by Brinker et al., published in 2021, failed to predict sentinel lymph node status in malignant melanoma to a clinically meaningful extent using deep learning-based image analysis [25]. In a 2020 study, Kulkarni et al. created a risk classifier that was significantly associated with the occurrence of recurrence in melanoma [26]. However, this score includes other factors for calculation, such as density and distribution of the immune cell infiltrate and nucleus morphology. Therefore, the impressive AUC values of 0.905 and 0.880 achieved in this study are not comparable to the results obtained here. Since other information besides the RGB image had to be included, tumor areas containing lymphocytes in addition to the tumor cells had to be available and the sections should not be too pigmented to allow detection of cellular components [26]. Another unique feature of our risk classifier is that it is a score that can be calculated with an image of only 103 kB to 622 kB in size. There is still a low availability of so-called whole-slide scanners, which can scan and digitize entire histological slides in high resolution in just a few minutes. Although this technology has been established for years, only a few pathological institutes have switched their routine settings to digital reporting, especially because of the high investment costs. Possibly in the coming years, the amount of memory and access to whole-slide scanners will no longer be limiting factors. Currently, a freely available, easy-to-use classifier operating on small data offers massive advantages when it comes to the question of validating that classifier in a large multicenter setting.

### 4.2. Limitations

The present study has several limitations. One potential point of criticism is the choice of deep learning tool. It is conceivable that an even better prediction of the prognosis could be made with different programs, although this was not investigated in this study. The focus of this research lies in the proof-of-concept, which shows that it is possible to make a prognosis prediction on the histological section with an as simple as possible AI application and as small as possible amount of data. Due to its straightforward transferability as well as its user-friendly interface, the publicly available Google’s Teachable Machine was chosen as a deep learning tool. Overfitting describes learning by memorization of the correct answers by the AI model instead of the establishment of a generally applicable assignment rule in the sense of generalization. Such an overfitting was evident in our trained models, even when repeated with reassigned image groups. It is possible that this overfitting could be minimized by various fine adjustments in the AI model, especially by adjusting the number of epochs. However, this was not further investigated in the present study. It is also conceivable that the pre-trained algorithm of the Teachable Machine is not suitable for this complex histological challenge and thus represents the limiting factor in model performance. Further limitations are that the training and test cohorts are retrospective evaluations and that the number of cases in the groups and especially the number of events included (39 deaths in the test cohort) is quite small. Another point of criticism is that all used sections originate from one and the same pathological institute. Possibly, the results show only limited transferability to other institutes, as a slightly different staining pattern in H&E staining may be evident here. In addition, the manual selection of the areas of interest by the pathologist offers the possibility of an influence. A trade-off must be made between large datasets and automated selection and manual selection and small data sets. Additionally, the use of similar images in the “dead” group of the training cohort may have restricted the learning curve of artificial intelligence. The melanoma treatment of the patients in our study was not examined. It is possible that changes in treatment regimens during the study period may have limited the predictive accuracy of AI prognosis. To obtain more meaningful results, a larger, prospectively designed, multicenter study would be necessary. One possibility for such studies in the future could be the use of so-called “swarm learning”. This newly described approach uses blockchain-based peer-to-peer networking to decentralize the use of machine learning [33].

Another problem with the method used here is the lack of explainability. A program that offers an explanatory approach implemented in the program would be desirable, so the black box of the AI could be illuminated. A study by Courtoil et al. from 2019 shows such a program that not only forecasts the prognosis of mesothelioma but can also show via a heat map analysis that the decision basis of the AI is to be found in the area of the tumor stroma [34].

## 5. Conclusions

Finally, the study presented here must be understood as proof-of-concept. It could be shown that prognostic information is contained in tiny image sections of a melanoma, which allows prognosis estimation. To establish a prognosis score that can be used in clinical practice, it must be clearly shown that such a score complements the current classification systems and may in the future be an alternative to invasive diagnostic methods, such as sentinel node biopsy or expensive gene-expression-based prognosis scores.

## Figures and Tables

**Figure 1 cancers-14-02243-f001:**
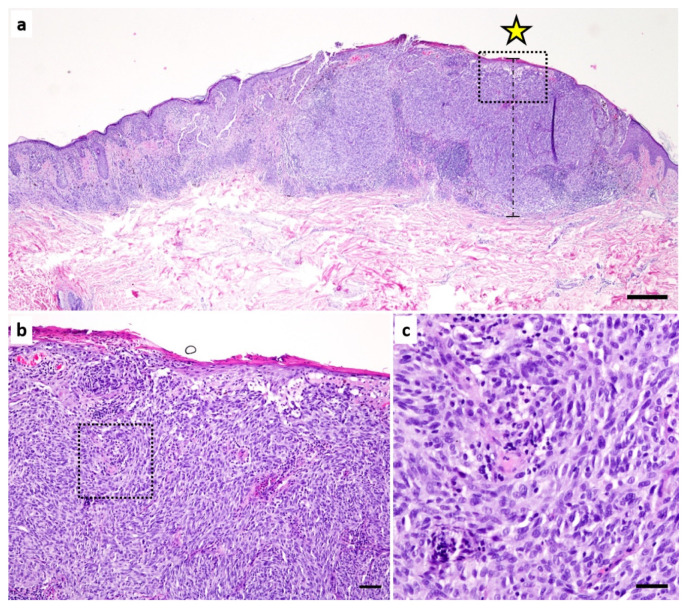
H&E section of a malignant melanoma. (**a**) overview with annotation (star) of the highest tumor thickness (Breslow). The scale is 500 µm. (**b**) Magnification of (**a**) (see square in (**a**)). The image represents one picture of the category “whole image”. The scale is 100 µm. (**c**) Magnification of (**b**) (see square in (**b**)). This image represents one picture of the category “area of interest”. The scale is 30 µm.

**Figure 2 cancers-14-02243-f002:**
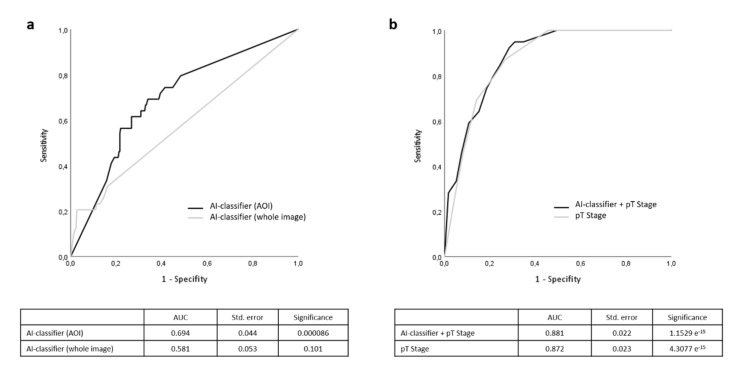
Average receiver operating characteristic (ROC) curves of overall survival prognosis. (**a**) Black line = AI-classifier with “area of interest” analysis. Gray line = AI-classifier with “whole image” analysis. (**b**) Black line = pT stage combined with AI-classifier (AOI). Gray line = pT stage (tumor thickness and presence of ulceration).

**Figure 3 cancers-14-02243-f003:**
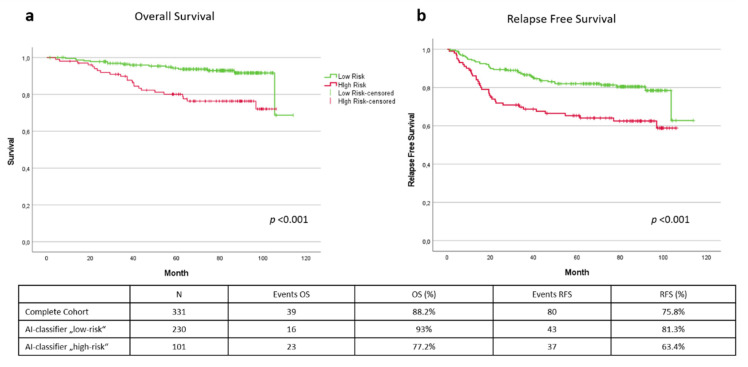
Kaplan–Meyer curve of overall survival (**a**) and relapse-free survival (**b**). Green line = AI-classifier “low risk”. Red line = AI-classifier “high risk”.

**Figure 4 cancers-14-02243-f004:**
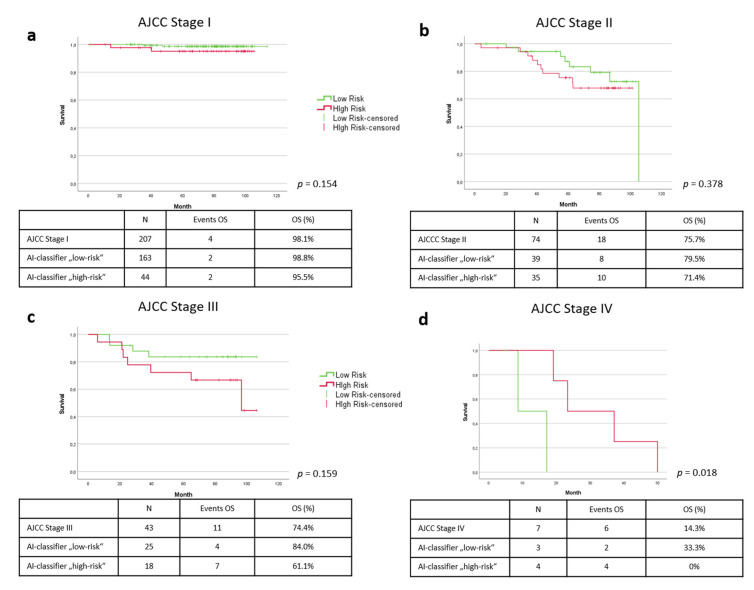
Kaplan–Meyer curves of overall survival in AJCC (2017) substages I (**a**), II (**b**), III (**c**) and IV (**d**). Green line = AI-classifier “low-risk”. Red line = AI-classifier “high-risk”.

**Table 1 cancers-14-02243-t001:** Demographics, tumor parameters, stage of disease (AJCC 2017), tumor subtype and survival of the cohort.

Demographics and Tumor Parameters	All (*n* = 831)	Training Cohort (*n* = 500)	Test Cohort (*n* = 331)
**Age at Diagnosis (years)**			
Min./Max.	7/93	9/93	7/91
Median (+IQR)	62 (49/72)	63 (50/73)	59 (48/71)
Mean value (±SD)	59.88 (±15.3)	61.06 (±15.0)	58.11 (±15.7)
**Sex (*n*, %)**			
Male (*n*, %)	462 (55.6%)	285 (57%)	177 (53.5%)
Female (*n*, %)	369 (44.4%)	215 (43%)	154 (46.5%)
**Primary tumor**			
Tumor thickness (Breslow, mm), Median (+IQR)	1.05 (0.5/2.4)	1.00 (0.45/2.2)	1.10 (0.55/2.5)
Ulceration (*n*, %)	177 (21.3%)	103 (20.6%)	74 (22.4%)
**Histologic subtype**			
Superficially spreading melanoma (SSM) (*n*, %)	493 (59.3%)	303 (60.6%)	190 (57.4%)
Nodular melanoma (NM) (*n*, %)	134 (16.1%)	75 (15.0%)	59 (17.8%)
Lentigo Maligna melanoma (LMM) (*n*, %)	76 (9.1%)	52 (10.4%)	24 (7.3%)
Acrolentiginous melanoma (ALM) (*n*, %)	50 (6.0%)	27 (5.4%)	23 (6.9%)
Others (*n*, %)	47 (5.7%)	27 (5.4%)	20 (6.0%)
Unknown (*n*, %)	29 (3.5%)	15 (3.0%)	14 (4.2%)
**Localisation**			
Head/neck (*n*, %)	147 (17.7%)	91 (18.2%)	56 (16.9%)
Trunk (*n*, %)	344 (41.4%)	224 (44.8%)	120 (36.3%)
Upper Extremities (*n*, %)	117 (14.1%)	67 (13.4%)	50 (15.1%)
Lower Extremities (*n*, %)	219 (26.4%)	116 (23.2%)	103 (31.1%)
Others/unknown (*n*, %)	4 (0.4%)	2 (0.4%)	2 (0.6%)
**Stage (AJCC 2017)**			
IA (*n*, %)	401 (48.3%)	248 (49.6%)	153 (46.2%)
IB (*n*, %)	133 (16.0%)	79 (15.8%)	54 (16.3%)
IIA (*n*, %)	80 (9.6%)	45 (9%)	35 (10.6%)
IIB (*n*, %)	60 (7.2%)	37 (7.4%)	23 (6.9%)
IIC (*n*, %)	35 (4.2%)	19 (3.8%)	16 (4.8%)
IIIA (*n*, %)	24 (2.9%)	14 (2.8%)	10 (3%)
IIIB (*n*, %)	23 (2.8%)	15 (3%)	8 (2.4%)
IIIC (*n*, %)	62 (7.5%)	37 (7.4%)	25 (7.6%)
IIID (*n*, %)	2 (0.2%)	2 (0.4%)	0
IV (*n*, %)	11 (1.3%)	4 (0.8%)	7 (2.1%)

## Data Availability

The trained model for the “whole image” analysis used in this study is available at https://teachablemachine.withgoogle.com/models/q3W4kP4zk/ (accessed on 3 January 2022). The trained model for the “area of interest” analysis used in this study is available at https://teachablemachine.withgoogle.com/models/EWFL98pti/ (accessed on 3 January 2022).

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
