# Peer review of "Development of an Image Analysis-Based Prognosis Score Using Google’s Teachable Machine in Melanoma"

_cancers, 2022, doi:10.3390/cancers14092243_

Round 1
Reviewer 1 Report
Dear Authors,
thank you for preparing this interesting and well-written article concerning deep-learning-based image analyses of melanoma histological sections.
First of all, indeed there is a necessity to use other forms of melanoma assessment than subjective assessment by a dermatologist, thus deep learning techniques may be the answer to better cancer diagnostics and prognostics. I really appreciated the criticism with which the authors approached the presented results (the last part of the discussion is devoted to the limitations of this given methodology).
With this being seed I have only some minor concerns for the authors:
- As in the results part some types of melanoma are being listed (e.g., superficial spreading melanoma, nodular melanoma, etc. page 4) I would recommend increasing a bit the introduction with a small part dedicated to the differences between those types of melanoma and in general to some basic statistics and information why melanoma is such a problem, especially nowadays (morbidity, mortality, resistance to therapies, etc.)
- The Authors like to repeat the word "however" and the sentence order associated with it, please correct it (especially pages 2, 7, 8)
- Please change all commas in numbers into dots (all manuscript, tables, and graphs)
- Please write all Latin words and expressions in italics (eg., versus, et al. )
- Please separate the number from the unit (eg. 3 ms, 103 kB, 500 µm)
- Please correct the spelling of the words "deep-learning-based" (page 1), "well-studied" (page 2), "gene-expression-based" (page 2)
Author Response
Manuscript ID: cancers-1690848
Response to Reviewer 1
Dear Reviewer,
Thank you for giving us the opportunity to submit a revised version of our paper "Development of an image analysis-based prognosis score using Google's Teachable Machine in melanoma." We highly appreciate the time and effort you have devoted to providing feedback on our manuscript as a reviewer. We are very grateful for the insightful comments and valuable improvements to our paper. The majority of the proposed changes were incorporated into the manuscript, the alterations are marked with tracked changes in the revised manuscript. Page and line numbers in this response refer to the revised manuscript.
Reviewers comments:
thank you for preparing this interesting and well-written article concerning deep-learning-based image analyses of melanoma histological sections.
First of all, indeed there is a necessity to use other forms of melanoma assessment than subjective assessment by a dermatologist, thus deep learning techniques may be the answer to better cancer diagnostics and prognostics. I really appreciated the criticism with which the authors approached the presented results (the last part of the discussion is devoted to the limitations of this given methodology).
Author response: Thank you very much!
- As in the results part some types of melanoma are being listed (e.g., superficial spreading melanoma, nodular melanoma, etc. page 4) I would recommend increasing a bit the introduction with a small part dedicated to the differences between those types of melanoma and in general to some basic statistics and information why melanoma is such a problem, especially nowadays (morbidity, mortality, resistance to therapies, etc.)
Author response: We have added a paragraph to the introduction which roughly describes the epidemiology of melanoma and the histological differentiation of subtypes according to the current WHO classification. (Page 2, line 53-68)
- The Authors like to repeat the word "however" and the sentence order associated with it, please correct it (especially pages 2, 7, 8)
Author response: We have changed a majority of the sentences that show the mentioned phrasing. (Page 2, line 83-85; page 2, line 98; page 7, line 717-718; page 7, line 737; page 8, line 771-773)
- Please change all commas in numbers into dots (all manuscript, tables, and graphs)
Author response: We have included the suggested changes in the manuscript. Commas in numbers have been changed to dots.
- Please write all Latin words and expressions in italics (eg., versus, et al. )
Author response: We have included the suggested changes in the manuscript. All Latin words and expressions are now written in italics.
- Please separate the number from the unit (eg. 3 ms, 103 kB, 500 µm)
Author response: We have included the suggested changes in the manuscript. Numbers are now separated from the unit.
- Please correct the spelling of the words "deep-learning-based" (page 1), "well-studied" (page 2), "gene-expression-based" (page 2)
Author response: The spelling is corrected accordingly.
Reviewer 2 Report
The authors have obtained and presented a very valuable dataset with a high potential for use with artificial intelligence algorithms.
Nevertheless, the choice of deep learning tool (Google's Teachable Machine) does not allow to reach good results and analyze obtained results. It is advisable to check a simpler neural model that is not pre-retained on complex objects like humans, cars, etc.
Since the number of episodes in certain melanoma stages is low and does not allow reaching adequate statistical significance, it is advisable to use cross-validation techniques to allow using all data in training. Or joining multiple melanoma stages.
Questions/comments:
1) Please clearly define train/validation and test datasets sizes, and usage principles and check their correct naming. Validation set - is accessible by training algorithm during training. Test set - data that have not been previously "seen" by the neural model. Use Google's Teachable Machine results after training to check the train/validation split.
2) Lines 119 - 123 and 128-133 are repeated.
3) Add a train/loss curve to check training quality and to make sure that there is no overfitting.
4) How many times training was repeated?
5) Please clarify how 'p' was evaluated.
6) How ground-truth classes "low risk" and "high risk" were defined? What is their relation to the prediction of overall survival?
Author Response
Manuscript ID: cancers-1690848
Response to Reviewer 2
Dear Reviewer,
Thank you for giving us the opportunity to submit a revised version of our paper "Development of an image analysis-based prognosis score using Google's Teachable Machine in melanoma." We highly appreciate the time and effort you have devoted to providing feedback on our manuscript as a reviewer. We are very grateful for the insightful comments and valuable improvements to our paper. The majority of the proposed changes were incorporated into the manuscript, the alterations are marked with tracked changes in the revised manuscript. Page and line numbers in this response refer to the revised manuscript.
Reviewers comments:
The authors have obtained and presented a very valuable dataset with a high potential for use with artificial intelligence algorithms.
Author response: Thank you very much!
Nevertheless, the choice of deep learning tool (Google's Teachable Machine) does not allow to reach good results and analyze obtained results. It is advisable to check a simpler neural model that is not pre-retained on complex objects like humans, cars, etc.
Author response: Thank you for the valuable comments on the study. We are aware that the choice of the deep learning tool is a possible point of criticism. Nevertheless, it cannot be assumed that the results obtained with this tool are inferior to those of a simpler neural network. This question has not been investigated in this study. The aim was not to create a new AI specifically focused on our dataset, but to provide forecast estimation using accessible tools. Google's Teachable Machine was deliberately chosen as the deep learning tool because it allows simple data processing and has a simple user interface. The study serves as a proof of concept that it is possible to predict prognostic factors with basic images and simple AI. In our opinion, the value of the study is therefore not affected by the choice of the deep learning platform. However, to disclose this limitation, a corresponding paragraph has been added in the discussion section/limitations. (Page 9, line 807-812)
Since the number of episodes in certain melanoma stages is low and does not allow reaching adequate statistical significance, it is advisable to use cross-validation techniques to allow using all data in training. Or joining multiple melanoma stages.
Author response: It is true that the numbers of patients in the individual tumor stages are partly low. However, the cohort was not assembled based on tumor stages. It is a broad group of all patients who received a diagnosis of malignant melanoma during the time of study coverage. The aim of the evaluation was not to classify an image to a certain tumor stage but to detect morphological characteristics by the AI which allow a general prognosis independent of the tumor stage. Therefore, we conclude that despite the limited number of episodes of individual stages, the use of these data for the training is reasonable.
- Please clearly define train/validation and test datasets sizes, and usage principles and check their correct naming. Validation set - is accessible by training algorithm during training. Test set - data that have not been previously "seen" by the neural model. Use Google's Teachable Machine results after training to check the train/validation split.
Author response: The use of the terms training-, validation- and test-cohort was indeed unclearly presented. The terms have now been defined in the methods section and adjusted accordingly throughout the manuscript. The Teachable Machine program does not provide a function for validation. This is integrated in the training function. Therefore, a special validation dataset has not been designated. A corresponding addition has also been inserted in the method section. (Page 3, line 158-167)
- Lines 119 - 123 and 128-133 are repeated.
Author response: The first paragraph (formerly line 119-123) has been deleted accordingly in the manuscript.
- Add a train/loss curve to check training quality and to make sure that there is no overfitting.
Author response: We have willingly included the training curves for accuracy and loss for both the training of the whole image group and the training of the area of interest group. These were created and output by Google's Teachable Machine. The curves were included as Supplementary Figure 1 to the manuscript. A corresponding reference has been inserted in the methods section. (Page 3, line 163-164)
- How many times training was repeated?
Author response: The training was conducted once for the whole image group and once for the area of interest group. A corresponding note explaining this has been added in the method section. (Page 3, line 162-163)
- Please clarify how 'p' was evaluated.
Author response: As already outlined in the Methods section, p values of Kaplan-Meier curves calculated using the log-rank (Mantel-Cox) test. The significance of the ROC curves is tested via the asymptotic significance with the null hypothesis: truth surface=0.5. Given the complexity of assessing the ROC curve standard error, an established statistical method for testing the significance of ROC curves is missing. P-Values of ROC-Curves were therefore calculated using the ROC-Analysis tool in SPSS. A corresponding note has been added to methods section. (Page 4, line 207)
- How ground-truth classes "low risk" and "high risk" were defined? What is their relation to the prediction of overall survival?
Author response: The group with the label "low risk" corresponds to the patients who were classified as "dead" by Google's teachable machine. The group with the label "high risk" corresponds to the patients who were evaluated as "alive". A corresponding paragraph explaining this has been added to the method section. (Page 3, line 167-170)
Reviewer 3 Report
The aim of the study presented is to correlate the images of the pigmented lesions of the skin to the prognosis of survival through the analysis of an artificial intelligence. Undoubtedly the topic is interesting and well developed by the authors. Unfortunately, some important biases are worth mentioning.
First, line 118, "the images in landscape 4:3 were randomly cut into a square format": who choose which part of image has to be cut and ignored?
Second, line 119 and 128-129, the two groups "alive" and "dead" were balanced through a division of images and a repeating use of some of these, is it correct? I guess it may limit the "learning curve" of the A.I. exposing it to a similar type of images.
Third, line 124, then correctly cited in "Study limitations", the "area of interest" of the single image was selected by a dermapathologist, so a selection bias will be unavoidable.
Fourth, line 149-151, the subtypes of melanoma submitted to A.I. in the training cohort respected the frequency of clinical diagnosis, but the scope of the training is to transfer informations regarding the melanoma-features and not the probability that a determined subtype will occur.
Fifth, perhaps the primary concern with this study is whether or not the 831 patients in the population study had the same treatments? Went through the same follow-up? The reason of death for melanoma was the spread of the disease or a limited adherence to the current therapy algorithm (from 2012-2015 to now something has changed...)?
Globally a well-written study, correctly presented as proof-of-concept, which can serve as a "trailblazer" for the development of more sophisticated and reliable decision-making algorithms.
I believe that an implementation of the section devoted to the study limitations is necessary.
Author Response
Manuscript ID: cancers-1690848
Response to Reviewer 3
Dear Reviewer,
Thank you for giving us the opportunity to submit a revised version of our paper "Development of an image analysis-based prognosis score using Google's Teachable Machine in melanoma." We highly appreciate the time and effort you have devoted to providing feedback on our manuscript as a reviewer. We are very grateful for the insightful comments and valuable improvements to our paper. The majority of the proposed changes were incorporated into the manuscript, the alterations are marked with tracked changes in the revised manuscript. Page and line numbers in this response refer to the revised manuscript.
Reviewers comments:
The aim of the study presented is to correlate the images of the pigmented lesions of the skin to the prognosis of survival through the analysis of an artificial intelligence. Undoubtedly the topic is interesting and well developed by the authors. Unfortunately, some important biases are worth mentioning.
Author response: Thank you very much!
- line 118, "the images in landscape 4:3 were randomly cut into a square format": who choose which part of image has to be cut and ignored?
Author response: The images were uploaded to Google's Teachable Machine in 4:3 format and cropped to a square image by the program. A manual selection of the crop was not done here. A corresponding paragraph was added in the method section. (Page 3, line 172-173)
- line 119 and 128-129, the two groups "alive" and "dead" were balanced through a division of images and a repeating use of some of these, is it correct? I guess it may limit the "learning curve" of the A.I. exposing it to a similar type of images.
Author response: That is correct. The groups were balanced by using similar images in the "dead" group. In fact, this may limit the learning curve of the AI. A sentence to reflect this has been added to the discussion under the limitations section. (Page 9, line 822-823)
- line 124, then correctly cited in "Study limitations", the "area of interest" of the single image was selected by a dermapathologist, so a selection bias will be unavoidable.
Author response: This is also correct. There is an unavoidable selection bias, since a compromise had to be found between large data sets without selection bias and small data sets with the disadvantage of manual selection. A corresponding paragraph which describes this limitation can already be found in the discussion. (Page 9, line 819-821)
- line 149-151, the subtypes of melanoma submitted to A.I. in the training cohort respected the frequency of clinical diagnosis, but the scope of the training is to transfer informations regarding the melanoma-features and not the probability that a determined subtype will occur.
Author response: The task of the AI was not to identify the subtype of melanoma in a specific image, but to identify superordinate structures that allow prediction of prognosis. The histological subtypes and other tumor parameters were analyzed in the total cohort, the training cohort and the test cohort only for the following reasons: 1. To show that it is a representative cohort of melanoma patients without special selection. 2. that it is a random distribution of groups with balanced tumor parameters.
- perhaps the primary concern with this study is whether or not the 831 patients in the population study had the same treatments? Went through the same follow-up? The reason of death for melanoma was the spread of the disease or a limited adherence to the current therapy algorithm (from 2012-2015 to now something has changed...)?
Author response: We can well understand the consideration. All patients included in the study had a follow-up of at least 2 years if not previously deceased. This is described in the methods section. How the individual patients were treated was not investigated in the study. It is correct that the AI may recognize structures that are associated with the treatment. It is possible that structures in the histological sections were detected which correlate with a response or non-response, e.g. to immunotherapy. However, due to the lack of explainability of the method, this question cannot be answered by the present study. A corresponding paragraph stating this limitation has been included in the discussion section. (Page 9, line 823-826)
- Globally a well-written study, correctly presented as proof-of-concept, which can serve as a "trailblazer" for the development of more sophisticated and reliable decision-making algorithms. I believe that an implementation of the section devoted to the study limitations is necessary.
Author response: A proof of concept study which can serve as a precursor for further research is exactly the intention of this work. In order to better define the study limitations, a separate subchapter was introduced in the discussion section.
Round 2
Reviewer 2 Report
Thank you for the corrections and the additional material. I agree that it is enough to prove the concept even if classification quality is not good enough. Nevertheless, a strong overfitting effect could be seen by looking at the training curves. That might show on an incorrectly trained model. I would strongly suggest retraining the model using different images from your dataset. No need to repeat validation and change all statistical calculations.
If the overfitting effect stays after retraining, that should be noted in the results section. It is preferable to add a possible explanation for why that happened. My guess is that model architecture is too complex for such images, and therefore Google Teaching Machine should not be used for such images.
Author Response
Manuscript ID: cancers-1690848
Revised response to Reviewer 2
Dear Reviewer,
Thank you for giving us the opportunity to resubmit a revised version of our paper "Development of an image analysis-based prognosis score using Google's Teachable Machine in melanoma". We highly appreciate the time and effort you spent as a reviewer to provide feedback on our manuscript again. The suggested changes have been incorporated into the manuscript, which has been revised again, and the changes are marked "tracked changes" in the manuscript. Page and line numbers in this statement refer to the revised manuscript.
Reviewers comments:
Thank you for the corrections and the additional material. I agree that it is enough to prove the concept even if classification quality is not good enough. Nevertheless, a strong overfitting effect could be seen by looking at the training curves. That might show on an incorrectly trained model. I would strongly suggest retraining the model using different images from your dataset. No need to repeat validation and change all statistical calculations.
Author response: As proposed, we performed a repeated training with newly assigned groups. The training curves collected here can be found in Supplementary Figure 2. A corresponding addendum has been inserted in the methods section. (Page 3, line 135-138)
If the overfitting effect stays after retraining, that should be noted in the results section. It is preferable to add a possible explanation for why that happened. My guess is that model architecture is too complex for such images, and therefore Google Teaching Machine should not be used for such images.
Author response: Since the overfitting was similarly seen in the repetition of the training, this was noted in the results section of the paper as you suggested (Page 4, line 173-176). The repeated training makes group selection bias unlikely. It is possible that overfitting could be improved by fine-tuning the teachable machine, but this is outside the scope of the study and was not investigated separately. It is also possible that the pre-trained model of the Teachable Machine is not suitable to answer such a complex histological question in full quality. A corresponding paragraph was inserted in the discussion section. (Page 10, line 314-322)